# Whole-Exome Sequencing and Analysis of the T Cell Receptor β and γ Repertoires in Rheumatoid Arthritis

**DOI:** 10.3390/diagnostics14050529

**Published:** 2024-03-01

**Authors:** Jooyoung Cho, Juwon Kim, Ju Sun Song, Young Uh, Jong-Han Lee, Hyang Sun Lee

**Affiliations:** 1Department of Laboratory Medicine, Yonsei University Wonju College of Medicine, Wonju 26426, Republic of Korea; purelove0927@yonsei.ac.kr (J.C.);; 2GC Genome, GC Labs, Yongin 16924, Republic of Korea; 3Division of Rheumatology, Department of Internal Medicine, Yonsei University Wonju College of Medicine, Wonju 26426, Republic of Korea

**Keywords:** whole-exome sequencing, T cell receptor repertoire, diversity, rheumatoid arthritis

## Abstract

This study investigated the potential genetic variants of rheumatoid arthritis (RA) using whole-exome sequencing (WES) and evaluated the disease course using T cell receptor (TCR) repertoire analysis. Fourteen patients with RA and five healthy controls (HCs) were enrolled. For the RA patient group, only treatment-naïve patients were recruited, and data were collected at baseline as well as at 6 and 12 months following the initiation of the disease-modifying antirheumatic drug (DMARD) treatment. Laboratory data and disease parameters were also collected. Genetic variants were detected using WES, and the diversity of the TCR repertoire was assessed using the Shannon–Wiener diversity index. While some variants were detected by WES, their clinical significance should be confirmed by further studies. The diversity of the TCR repertoire in the RA group was lower than that in the HCs; however, after DMARD treatment, it increased significantly. The diversity was negatively correlated with the laboratory findings and disease measures with statistical significance. Variants with a potential for RA pathogenesis were identified, and the clinical significance of the TCR repertoire was evaluated in Korean patients with RA. Further studies are required to confirm the findings of the present study.

## 1. Introduction

Rheumatoid arthritis (RA) is a systemic autoimmune disease characterized by chronic inflammatory synovitis, joint and bony destruction, and multi-organ manifestations, resulting in morbidity and mortality [1,2]. RA has many complex etiologies involving genetic, immunological, and acquired factors [2]. In addition, environmental factors, such as smoking, respiratory diseases, and changes in the gut microbiome, may increase the risk of or cause epigenetic modifications in susceptible patients, which may lead to RA development [3,4]. For these reasons, identifying target genes for RA is very important for preventing and/or treating patients with RA, which can also be applied to further understand the pathogenesis of RA and potential targets of therapy [5,6].

RA is an autoimmune disease and many studies have been conducted on autoantibodies; however, a consensus on the crucial autoantigens or genetic and environmental factors that trigger the autoimmune process has not yet been established [7]. The cause of RA is not yet clearly known because of the complex traits of various factors; however, it is widely established that genetic factors account for approximately 50% of RA cases [8,9]. For many decades, many studies have been conducted to identify genetic variants in RA and identify treatment targets. Owing to the development of genome-wide association studies (GWAS), many genetic variants related to RA have been discovered [10]. Genetic factors are important not only for preventive activities in individuals with a high probability of RA but also for the treatment process, response, severity, and prognosis [11]. Current conventional and biological treatments sometimes fail or show only a partial response. If the genetic marker is well identified, it is expected that the response to treatment and prognosis can be improved while minimizing toxicity through targeted treatment [7]. Although many studies have been conducted on the susceptibility to RA development using single nucleotide polymorphisms (SNPs), which have attempted to identify disease-specific genes or polymorphisms, these SNPs alone cannot accurately explain the etiology because the RA-related genetic background is complex [12]. GWAS have suggested genetic loci for susceptibility to RA [13,14,15,16]. With the advent of next-generation sequencing (NGS), some potential genetic variants have been analyzed [17]. Meanwhile, abnormal and pathogenic T cell responses that evade normal immune functions could be considered one of the mechanisms of RA development [3], and the T cell receptor (TCR) repertoire has been studied in patients with RA [18]. However, to date, there have been only a limited number of related studies, with no published research on Korean patients.

For these reasons, this study aimed to investigate the potential genetic variants for RA development in Korean patients using whole-exome sequencing (WES) and the pathogenesis of disease course using TCR β and γ repertoire (TRB and TRG) analysis. This study aimed to provide basic data that can be applied to Korean patients by analyzing and evaluating the results at a molecular genetic level.

## 2. Materials and Methods

### 2.1. Study Design and Data Collection

From December 2020 to February 2022, study participants were recruited from patients who visited the outpatient department (OPD) of the Rheumatology of Internal Medicine at Wonju Severance Christian Hospital, a tertiary university-affiliated hospital located in Wonju, South Korea. The inclusion criterion of this study was treatment-naïve patients who were first diagnosed with RA in our hospital and who had not yet started the disease-modifying antirheumatic drug (DMARD) treatment. A total of 17 patients were enrolled and scheduled to visit the OPD at various time points: the first time before initiating DMARD treatment, and then 6 and 12 months after treatment. However, 3 patients were lost to follow-up during the study period, and finally 14 patients remained; their blood samples were collected for up to 12 months and used for data analysis. Five healthy controls (HCs) without diagnosed RA were also enrolled for comparison purposes, requiring only one visit. At each visit, blood samples were collected in two K_2_-ethylenediaminetetraacetic acid (EDTA) tubes, one for measuring the erythrocyte sedimentation rate (ESR) and the other for genomic testing, as well as one serum separating tube for measuring other laboratory data.

Baseline characteristics were collected as follows: age, sex, medical history of hypertension and diabetes mellitus, height, body weight, and the duration before the first visit for clinical information; tender joint count of 44/28 joints (TJC44/28); swollen joint count of 44/28 joints (SJC44/28); disease activity score in 28 joints (DAS28); health assessment questionnaire (HAQ); visual analog scale (VAS) score; simplified disease activity index (SDAI); clinical disease activity index (CDAI); and fulfillment of the 1987 American College of Rheumatology (ACR) criteria (whether ≥4 out of 7 cases are met) [19] and the 2010 ACR/European Alliance of Associations for Rheumatology (EULAR) criteria (whether a score of ≥6 out of 10 was met) [20] for clinical classification or scoring system.

Laboratory data were collected on the following: antinuclear antibody (ANA) determined by a QUANTA Lite^®^ ANA assay (Inova Diagnostics, Inc., San Diego, CA, USA), rheumatoid factor (RF) and C-reactive protein (CRP) determined by the Cobas c 702 automated analyzer (Roche Diagnostics, Rotkreuz, Switzerland), anti-cyclic citrullinated peptide (ACCP) determined by a QUANTA^®^ Flash CCP3 assay (Inova Diagnostics), and ESR determined by the TEST-1 analyzer (SIRE Analytical Systems, Udine, Italy).

This study was approved by the Institutional Review Board (IRB) of Wonju Severance Christian Hospital (IRB No. CR320086). All the participants voluntarily participated in the study and provided written informed consent.

### 2.2. Genomic Analysis

Peripheral blood mononuclear cells were isolated from EDTA-treated whole blood by Ficoll–Hypaque density gradient centrifugation. The protocols for WES and TRB/TRG were as follows.

For WES, genomic deoxyribonucleic acid (DNA) was extracted using the ChemagicTM Magnetic Separation Module I method (PerkinElmer Chemagen, Baesweiler, Germany) with a DNA blood 200 μL kit. The MGIEasy Exome Capture V5 Probe Set (MGI Tech Co., Ltd., Shenzhen, China) was used for library preparation, and sequencing was performed on the MGI DNBSEQ-G400 platform (MGI Tech Co., Ltd.) to generate 2 × 100 bp paired-end reads. DNA sequence reads were aligned to the reference sequence based on the public human genome build GRCh37/UCSC hg19. Alignments were performed with BWA-mem (version 0.7.17), duplicate reads were marked with biobambam2 and base quality recalibration, variant calling was performed with the Genome Analysis Tool kit (GATK, version 4.1.8), and annotation was performed with VEP101 (Variant Effect Predictor 101) and dbNSFP v4.1 [21].

For the TRB and TRG analysis, genomic DNA was extracted using the Chemagic^TM^ DNA Blood 200 kit (Chemagen, Baesweiler, Germany). For TCR repertoire sequencing, the Lymphotrack^®^ TRB and TRG assays (Cat No. 72270009 and 72270019, respectively) (Invivoscribe Inc., San Diego, CA, USA) were used for NGS. These assays amplify the DNA between the primers that target the conserved V and J regions of the T cell receptor genes, TRB and TRG [22,23].

Polymerase chain reaction amplicons were purified and quantified using a Qubit fluorometer (Thermo Fisher Scientific, Waltham, MA, USA). Equimolar amounts of libraries were pooled and loaded onto a flow cell using a MiSeq Reagent kit v2 (500 cycles) (Illumina Inc., San Diego, CA, USA) and sequenced on a MiSeq instrument (Illumina).

### 2.3. Statistical Analysis

The numerical data results were checked for normality using the Kolmogorov–Smirnov test, which determined that a *p*-value greater than 0.05 had normality. For the parametric data, the results are presented as the means ± standard deviations (SDs), and the Student’s *t*-test was used for comparison. For the non-parametric data, there results are presented as the median and interquartile range (IQR), and the Mann–Whitney U test was used for comparison. The results of the categorical data are presented as frequencies and percentages (%), and the χ^2^ test was used for comparison.

For patients with RA, the results at baseline (treatment-naïve) were compared to those at the 6 and 12 month follow-up. To compare the DAS28, DAS28-ESR (=0.70 × ln [ESR]) and DAS28-CRP (=0.36 × ln [CRP + 1] + 0.96) [24] were calculated. All three VAS scores (pain VAS, global VAS, and physician VAS) were evaluated, but all showed almost the same value; therefore, only the pain VAS scores were used for comparison. All the comparisons, except for the analysis of variance, were made individually for each subgroup.

All the statistical analyses were performed using Analyse-it version 6.15.4 (Analyse-it Software, Ltd., Leeds, UK), an add-on program to Microsoft Excel 2019 (Microsoft Corp., Redmond, WA, USA), and SPSS (version 25.0; IBM Corp., Armonk, NY, USA). Statistical significance was set at a *p*-value of < 0.05.

## 3. Results

### 3.1. Baseline Characteristics

The baseline characteristics of the 14 patients with RA and 5 HCs are presented in Table 1. The median age of the patients with RA was 57 years (IQR, 45–69), and 12 (85.7%) of them were female. There was no statistically significant difference in the proportions of patients with hypertension, diabetes mellitus, or obesity between the patients with RA and the HCs. When comparing the RA criteria, 10 patients (71.4%) satisfied the 1987 ACR criteria, whereas all 14 patients (100%) satisfied the 2010 ACR/EULAR criteria. Laboratory findings showed that the proportions of people positive for ANA, RF, and ACCP were significantly higher in the patients with RA than in the HCs, and the median ESR and CRP values were also higher in the patients with RA than in the HCs.

### 3.2. Genomic Variants

Using WES, disease-related variants that have been reported to be associated with RA among several SNPs were detected in four (28.6%) of the patients with RA. Table 2 lists the variants detected using WES that are likely related to RA pathogenesis. Some of the patients with RA had these variants, and one patient even had multiple variants. However, these SNPs were not found in the HCs, leading us to believe that these variants may be associated with RA pathogenesis. Most of the variants were of uncertain significance (VUS), but there was one case of a pathogenic variant (PV).

### 3.3. Changes in Clinical Features after DMARD Treatment

Table 3 displays the results of comparing the clinical features of the treatment-naïve patients with RA before starting DMARD treatment with their clinical features at 6 and 12 months following the initiation of the DMARD treatment. For the DMARD treatment, mainly 10 to 15 mg of methotrexate was used per week, and 1 g of sulfasalazine per day was added for some patients. Notably, there were no non-responders or poor responders to the DMARD treatment. Compared to the baseline (before treatment), the laboratory findings after 12 months of treatment decreased significantly. There were also changes in the number of joints; the TJC44, SJC44, TJC28, and SJC28 counts at 6 months decreased significantly compared to the baseline. These facts are also reflected in the disease measures; the DAS28, HQA, VAS, SDAI, and CDAI scores decreased significantly at 6 months, and even at 12 months they remained significantly decreased compared to the baseline.

### 3.4. Changes in TCR Diversity after DMARD Treatment

The TRB and TRG diversities were assessed using the Shannon–Wiener diversity index. The formula is as follows:$$\mathrm{Shannon}–\mathrm{Wiener}\;\mathrm{diversity}\;\mathrm{index}=\sum_{i=0}^{R}p_{i}{ln}_{p_{i}}$$
where *R* is the number of TCR CDR3 amino acid clones, and *p_i_* is the frequency of the *i*th clonotype [25]. In general, the values of the Shannon–Weiner diversity index are usually between 1.5 and 3.5 and are known to be rare in excess of 4.5 [26]. However, in previous RA studies, the values of the Shannon–Wiener diversity indices were very high and exceeded 4.5 [25,27].

The results of the comparison of TCR diversity between the HCs and patients with RA at baseline as well as at 6 and 12 months after starting DMARD treatment are illustrated in Figure 1. For TRB diversity, the Shannon–Wiener diversity indices in the HCs and patients with RA at baseline as well as at 6 months and 12 months after treatment were 0.69 (IQR, 0.63–0.74), 0.55 (IQR, 0.41–0.58), 0.56 (IQR, 0.43–0.73), and 0.96 (IQR, 0.66–1.52), respectively (Figure 1A). For TRG diversity, the Shannon–Wiener diversity indices in the HCs and patients with RA at baseline as well as at 6 months and 12 months after treatment were 0.42 (IQR, 0.37–0.48), 0.29 (IQR, 0.26–0.31), 0.32 (IQR, 0.24–0.43), and 0.51 (IQR, 0.35–0.76), respectively. The TRB and TRG diversities of the patients with RA after 12 months of treatment were much higher than when they were treatment-naϊve as well as after 6 months of treatment.

### 3.5. Relationship between TCR Diversity and RA-Related Factors

The relationship between TCR diversity and disease-related factors in RA was analyzed using Spearman’s rank correlation analysis. The numerical relationships between the TCR diversity and RA-related factors were considered, but the disease status (patients with RA or HCs) was not considered. Therefore, the data of the patients with RA and HCs were combined and analyzed together. For the laboratory findings, data from the patients with RA and HCs were included and analyzed, and only the ESR showed a significant correlation with both TRB and TRG. Disease measures showed significant correlations with all the items evaluated, especially with the DAS28-ESR, SDAI, and CDAI, with the correlation coefficient exceeding 0.5 (Table 4).

Additionally, the relationship between the TCR (TRB and TRG) diversity (Shannon–Wiener diversity indices) and the value of each RA-related factor was obtained through linear regression. Figure 2 shows a linear regression diagram between the TCR diversity and laboratory values, with the thick line representing the regression line and the dotted line representing the 95% confidence interval. In Table 5, the slope and 95% confidence interval of the linear regression equation between the Shannon–Wiener diversity and RA-related factors are analyzed and presented using data from the patients with RA and HCs. Because the analytical measuring interval (AMI) of each test item varies for laboratory items and the scale of each scoring varies for the disease measures, it is difficult to uniformly compare using only the slope and intercept. However, it can be seen that, except for ACCP, compared to the TCR diversity, the slope is negative; therefore, it can be concluded that the improvement in the laboratory findings and disease measures (decreases in numeric values) is related to the increase in the TCR diversity.

## 4. Discussion

This study aimed to find possible variations for the onset of RA in Korean treatment-naϊve patients and to evaluate the usefulness of TCR repertoire analysis as a factor related to the disease course and the response to DMARD treatment. Of the 14 patients with RA who were finally enrolled in this study, all satisfied the 2010 ACR/EULAR criteria, but only 10 patients (71.4%) met the criteria for diagnosing RA according to the 1987 ACR criteria, which is known to have lower sensitivity for early RA diagnosis than the 2010 ACR/EULAR [20].

Based on the WES results, variants expected to be associated with RA were detected in 4 of 14 patients with RA. In the Janus kinase 3 (*JAK3*) gene, the variant c.1333C>T, p.Arg445Ter was detected, which has been reported as a pathogenic variant in severe combined immunodeficiency [28] but has not yet been reported in RA. However, its association with RA should be considered because JAK3 inhibitors may be used for RA treatment [29]; follow-up studies on this are required. Peptidyl arginine deiminase type IV (PADI4) is a gene that converts arginine residues into citrulline residues in the presence of calcium ions; mutations in this gene are known to overproduce citrulline and cause a loss of tolerance, thereby increasing vulnerability to RA [30]. Tumor necrosis factor ligand superfamily member 18 (TNFSF18) is known to activate T cells and B cells in connection with cell signaling [31], and tumor necrosis factor receptor-associated factor (TRAF) is known to be associated with TNF and interleukin (IL)-1 and to cause inflammation [32]. NF-κB is known to be activated in the synovium of patients with RA to regulate the genes contributing to inflammation, such as TNF, IL-6, and IL-8, and treatment therapy targeting this gene is under study [5]. However, it does not match the SNP found in the European [14] and Japanese studies [33]; the clinical significance of the SNP shown in this study is somewhat limited, and it is necessary to clarify this or to further discover other SNPs through follow-up studies on Korean patients. Of course, it will be helpful to develop a drug or treatment agent that targets gene mutations, as they affect the activity of certain proteins; however, more GWAS will be required as human genetic patterns are complex, making it difficult to set and establish targets and produce effective therapeutic effects [34].

Genetic and specific environmental factors combine to trigger RA, which activates an immune reaction and causes loss of immune tolerance; this also increases the number and binding affinity of autoantibodies [1]. The disease progression of RA occurs through epigenetic remodeling [1]. In the immune response, T cells and B cells are activated, resulting in an increase in inflammatory cytokines, production of matrix metalloproteinases, and induction and activation of osteoclasts; this results in bony destruction [3]. Because the aberrant expression of CD4+ T cells plays an important role in the pathogenesis of RA, the identification and quantitative comparison of CD4+ T cells are being studied in relation to the etiology of RA for early diagnosis and as a possible indicator of the disease course or response to treatment [27]. A previous study reported that the TCR repertoire helps in the early diagnosis of RA [18]. The TCR repertoire in patients with RA is known to be less diverse than in healthy individuals; this is related to the disease activity [25,27].

In this study, a TCR repertoire analysis was performed. Treatment-naïve patients without prior DMARD treatment were enrolled, and the data at baseline (before treatment) as well as at 6 and 12 months after the initiation of DMARD treatment were compared. The baseline laboratory findings of the patients with RA were significantly different from those of the HCs. The TCR (TRB and TRG) diversities of the patients with RA were lower than those of the HCs. However, the TCR diversity in the patients with RA increased after the initiation of DMARD treatment. Laboratory values, affected joint counts, and disease measures were significantly decreased. As the symptoms and disease-related factors improved according to the treatment response, TCR diversity increased. According to previous studies, as the progression or severity of a disease increases, the clinical expansion of T cells occurs, and their diversity decreases [35]. The diversity of the TCR repertoire may change according to treatment; monitoring changes in the TCR repertoire is expected to play an important role in monitoring the disease course, evaluating the response to treatment, confirming recurrence, and predicting the prognosis of patients with RA [36]. Correlations between the TCR repertoire diversity and disease-related factors were also evaluated. The factors were negatively correlated with TCR diversity; this is in line with the result of a previous RA cohort study [27]. The DAS, SDAI, and CDAI scores were more strongly correlated with TCR diversity than the laboratory findings. The relationships between TCR diversity and RA-related factors were obtained using linear regression. However, because the reference value and AMI of each item are different, and the scale of disease measures is different, correlations should not be determined using only the slope values of the equations.

This study has several limitations. First, the number of enrolled patients with RA was small. This is because only treatment-naïve patients without prior treatment who were diagnosed with RA at our hospital were enrolled. Furthermore, age- or gender-matched controls were not collected. We only recruited those who voluntarily visited the OPD through a notice of recruitment. Further studies with larger sample numbers and age- or gender-matched controls are required. Secondly, by using WES, the variants detected in this study have no definite associations with previous reports. Third, this study focused only on TCR diversity according to DMARD treatment. Follow-up studies with a larger number of patients with RA will be able to clarify the clinical significance of the variants detected by WES in this study, detect other variants, and increase the diagnostic and predictive value of TRB and TRG diversity as a marker for the diagnosis, treatment, and prognosis of RA.

## 5. Conclusions

This study investigated potential genetic variants for RA development in Korean patients using WES and evaluated the clinical significance of the TCR repertoire by analyzing TCR/TRB diversity. Some potential variants were detected using WES; however, further studies are required to confirm their clinical significance. This study is meaningful in that the Korean patients showed similar results; the disease-related factors and TCR diversities showed negative correlations and linear relationships because the correlation coefficient was high and statistically significant. This study provides basic data that can be applied to Korean patients by analyzing and evaluating the results at a molecular genetic level.

## Figures and Tables

**Figure 1 diagnostics-14-00529-f001:**
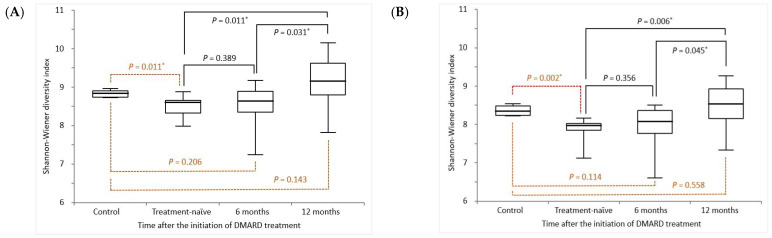
Comparison of the (**A**) TRB and (**B**) TRG diversity between the healthy controls and patients with RA at various time points following the initiation of DMARD treatment. TRB, T cell receptor β repertoires; TRG, T cell receptor γ repertoires; DMARD, disease-modifying antirheumatic drug. * *p* < 0.05 represents statistical significance. Black solid lines represent the *p* values between patients after DMARD treatment and treatment-naϊve patients. And, brown dotted lines represent the *p* values between patients after DMARD treatment and healthy controls.

**Figure 2 diagnostics-14-00529-f002:**
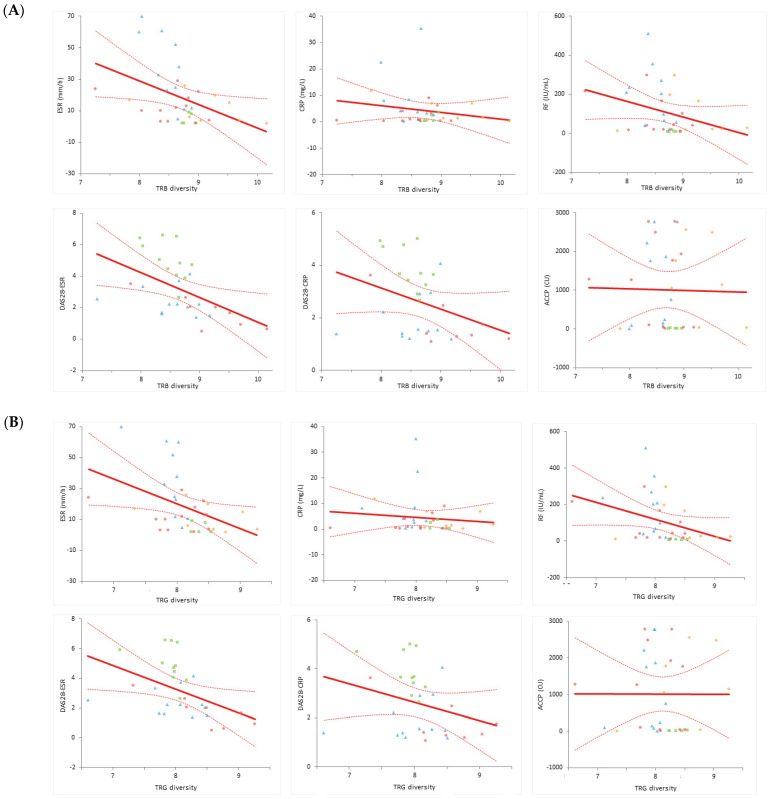
Linear regression between the (**A**) TRB and (**B**) TRG diversity and laboratory findings. TRB, T cell receptor β repertoires; TRG, T cell receptor γ repertoires; ESR, erythrocyte sedimentation rate; CRP, C-reactive protein; RF, rheumatoid factor; DAS28, disease activity score in 28 joints; ACCP, anti-cyclic citrullinated peptide. A total of *n* = 47 data (data of 14 patients with RA before and after DMARD treatment, as well as at 6 and 12 months, and 5 healthy controls’ data) were used for analysis. ■, healthy controls; ▲, treatment-naϊve patients; ●, patients with RA after 6 months of treatment; ◆, patients with RA after 12 months of treatment.

**Table 1 diagnostics-14-00529-t001:** Baseline characteristics of the study participants.

Baseline Variables	Unit	Feature	Patients with RA (*n* = 14)	HCs (*n* = 5)	*p*-Value
Age	Years	Median (IQR)	57 (45–69)	38 (31–48)	0.034 *
Female sex		No. (%)	12 (85.7)	3 (60.0)	0.046 *
Medical history of					
Hypertension		No. (%)	3 (21.4)	1 (20.0)	0.946
Diabetes mellitus		No. (%)	3 (21.4)	0 (0.0)	0.259
Obesity (BMI > 25 kg/m^2^)		No. (%)	4 (28.6)	1 (20.0)	0.709
Duration before the first visit	Months	Median (IQR)	8 (3–12)	-	-
RA criteria fulfillment					
ACR 1987		No. (%)	10 (71.4)	-	-
ACR/EULAR 2010		No. (%)	14 (100.0)	-	-
Disease-related variants		No. (%)	4 (28.6)	-	-
Laboratory findings					
ANA-positive		No. (%)	12 (85.7)	0 (0.0)	<0.001 *
RF-positive		No. (%)	13 (92.9)	0 (0.0)	<0.001 *
ACCP-positive		No. (%)	13 (92.9)	0 (0.0)	<0.001 *
ESR	mm/h	Median (IQR)	29 (12–53)	5 (2–8)	<0.001 *
CRP	mg/L	Median (IQR)	3.7 (2.2–8.2)	1.3 (0.3–2.8)	0.019 *

RA, rheumatoid arthritis; HC, healthy control; IQR, interquartile range; BMI, body mass index; ACR, American College of Rheumatology; EULAR, European Alliance of Associations for Rheumatology; ANA, antinuclear antibody; RF, rheumatoid factor; ACCP, anti-cyclic citrullinated peptide; ESR, erythrocyte sedimentation rate; CRP, C-reactive protein. * *p* < 0.05 represents statistical significance.

**Table 2 diagnostics-14-00529-t002:** Variants likely to be associated with rheumatoid arthritis detected using whole-exome sequencing.

Patient	Gene	DNA Change	AA Change	Zygosity	Class
1	JAK3	1333C>T	Arg445Ter	Hetero	PV
	PADI4	1861G>C	Glu621Gln	Hetero	VUS
	TNFSF18	167T>C	Met56Th	Hetero	VUS
	TRAF1	385C>T	Arg129Trp	Hetero	VUS
2	NFKB1	2708A>G	His903Arg	Hetero	VUS
3	TNFSF18	93G>A	Met31Ile	Hetero	VUS
4	TNFSF18	94C>T	Pro32Ser	Hetero	VUS

DNA, deoxyribonucleic acid; AA, amino acid; JAK3, Janus kinase 3; PV, pathogenic variant; PADI4, peptidyl arginine deiminase IV; VUS, variant of uncertain significance; TNFSF18, tumor necrosis factor ligand superfamily member 18; TRAF1, tumor necrosis factor receptor-associated factor 1; NFKB1, nuclear factor κB subunit1.

**Table 3 diagnostics-14-00529-t003:** Comparison of the clinical features of patients with rheumatoid arthritis before and after DMARD treatment.

Variables	Unit	Feature	Baseline	Time after DMARD Treatment		*p*-Value	
				6 Months	12 Months	Baseline vs. 6 Months	Baseline vs. 12 Months
Laboratory findings							
ESR	mm/h	Median (IQR)	29 (12–53)	11 (3–22)	10 (5–20)	0.006 *	0.005 *
CRP	mg/L	Median (IQR)	3.7 (2.2–8.2)	1.6 (0.7–7.0)	0.9 (0.4–3.5)	0.148	0.029 *
RF titer	IU/mL	Median (IQR)	148.5 (55.7–239.8)	42.0 (19.0–107.3)	33.5 (22.8–99.1)	0.051	0.033 *
ACCP titer	CU	Median (IQR)	1266.2 (37.8–1970.2)	947.8 (36.6–1831.8)	801.4 (150.6–1905.1)	0.963	0.748
Joint counts							
TJC44	no.	Median (IQR)	4 (3–9)	1 (0–3)	1 (0–2)	0.003 *	<0.001 *
SJC44	no.	Median (IQR)	3 (2–5)	0 (0–1)	0 (0–1)	<0.001 *	<0.001 *
TJC28	no.	Median (IQR)	3 (2–5)	0 (0–1)	0 (0–1)	<0.001 *	<0.001 *
SJC28	no.	Median (IQR)	2 (1–3)	0 (0–1)	0 (0–1)	<0.001 *	<0.001 *
Disease measures							
DAS28-ESR		Mean ± SD	4.75 ± 1.26	2.52 ± 0.89	2.16 ± 1.21	<0.001 *	<0.001 *
DAS28-CRP		Mean ± SD	3.68 ± 0.89	2.03 ± 0.90	1.98 ± 0.85	<0.001 *	<0.001 *
HAQ score		Mean ± SD	1.18 ± 0.65	0.64 ± 0.32	0.53 ± 0.35	0.013 *	0.008 *
Pain VAS score		Mean ± SD	6.71 ± 1.98	2.14 ± 1.17	1.89 ± 1.18	<0.001 *	<0.001 *
SDAI		Mean ± SD	16.54 ± 8.22	5.43 ± 3.62	3.68 ± 2.46	<0.001 *	<0.001 *
CDAI		Mean ± SD	16.43 ± 8.20	5.23 ± 3.47	4.76 ± 4.25	<0.001 *	<0.001 *

DMARD, disease-modifying antirheumatic drugs; ESR, erythrocyte sedimentation rate; IQR, interquartile range; CRP, C-reactive protein; RF, rheumatoid factor; ACCP, anti-cyclic citrullinated peptide; TJC44/28, tender joint count of 44/28; SJC44/28, swollen joint count of 44/28; DAS28, disease activity score in 28 joints; SD, standard deviation; HAQ, health assessment questionnaire; VAS, visual analog scale; SDAI, simplified disease activity index; CDAI, clinical disease activity index. * *p* < 0.05 represents statistical significance.

**Table 4 diagnostics-14-00529-t004:** Correlations between TRB/TRG diversity and disease-related factors.

	TRB		TRG	
	Spearman’s *r* (95% CI)	*p*-Value	Spearman’s *r* (95% CI)	*p*-Value
Laboratory findings				
ESR	−0.435 (−0.689 to −0.084)	0.015 *	−0.378 (−0.652 to −0.017)	0.036 *
CRP	−0.163 (−0.489 to 0.213)	0.380	−0.063 (−0.417 to 0.308)	0.737
RF titer	−0.267 (−0.575 to 0.107)	0.146	−0.277 (−0.583 to 0.096)	0.131
ACCP titer	0.076 (−0.488 to 0.226)	0.684	−0.019 (−0.380 to 0.347)	0.919
Disease measures				
DAS28-ESR	−0.580 (−0.780 to −0.274)	<0.001 *	−0.575 (−0.777 to −0.268)	<0.001 *
DAS28-CRP	−0.389 (−0.660 to −0.029)	0.031 *	−0.358 (−0.638 to −0.007)	0.048 *
HAQ score	−0.382 (−0.655 to −0.020)	0.034 *	−0.337 (−0.624 to 0.031)	0.064
Pain VAS score	−0.498 (−0.729 to −0.163)	0.004 *	−0.481 (−0.719 to −0.143)	0.006 *
SDAI	−0.552 (−0.763 to −0.236)	0.001 *	−0.576 (−0.777 to −0.268)	<0.001 *
CDAI	−0.561 (−0.768 to −0.248)	0.001 *	−0.587 (−0.784 to −0.284)	<0.001 *

TRB, T cell receptor β repertoires; TRG, T cell receptor γ repertoires; CI, confidence interval; ESR, erythrocyte sedimentation rate; CRP, C-reactive protein; RF, rheumatoid factor; ACCP, anti-cyclic citrullinated peptide; DAS28, disease activity score in 28 joints; HAQ, health assessment questionnaire; VAS, visual analog scale; SDAI, simplified disease activity index; CDAI, clinical disease activity index. * *p* < 0.05 represents statistical significance. A total of *n* = 47 data (data of 14 patients with RA before and after DMARD treatment, as well as at 6 and 12 months, and 5 healthy controls’ data) were used for analysis.

**Table 5 diagnostics-14-00529-t005:** Results of the linear regression between TCR diversity and disease-related factors.

		Slope	(95% CI)	Intercept	(95% CI)
TRB	Laboratory findings				
	ESR (mm/h)	−14.89	(−25.71 to −4.08)	147.79	(53.54 to 242.00)
	CRP (mg/L)	−2.52	(−7.04 to 2.01)	26.20	(−13.22 to 65.62)
	RF (IU/mL)	−78.84	(−156.60 to −1.04)	793.42	(115.55 to 1471.28)
	ACCP (CU)	−41.09	(−756.07 to 673.89)	1368.08	(−4861.37 to 7597.43)
	Disease measures				
	DAS28-ESR	−1.57	(−2.59 to −0.55)	16.78	(7.89 to 25.66)
	DAS28-CRP	−0.81	(−1.62 to 0.01)	9.58	(2.50 to 16.66)
	HAQ score	−0.64	(−1.14 to −0.13)	6.39	(1.99 to 10.80)
	Pain VAS score	−1.99	(−3.84 to −0.16)	20.99	(4.99 to 36.99)
	SDAI	−6.42	(−11.85 to −0.98)	64.68	(17.54 to 111.93)
	CDAI	−6.40	(−11.81 to −0.99)	64.42	(17.39 to 111.45)
TRG	Laboratory findings				
	ESR (mm/h)	−16.06	(−27.64 to −4.48)	148.62	(54.42 to 242.82)
	CRP (mg/L)	−1.62	(−6.52 to 3.29)	17.43	(−22.50 to 57.35)
	RF (IU/mL)	−94.32	(−176.45 to −12.20)	873.51	(205.61 to 1541.40)
	ACCP (CU)	−2.10	(−768.76 to 764.55)	1027.77	(−5207.48 to 7263.02)
	Disease measures				
	DAS28-ESR	−1.60	(−2.74 to −0.46)	16.07	(6.84 to 25.30)
	DAS28-CRP	−0.75	(−1.66 to 0.15)	8.67	(1.34 to 16.00)
	HAQ score	−0.68	(−1.23 to −0.12)	0.64	(1.85 to 10.85)
	Pain VAS score	−2.12	(−4.14 to −0.10)	20.79	(4.45 to 37.14)
	SDAI	−6.15	(−12.22 to −0.08)	58.68	(9.55 to 107.81)
	CDAI	−6.15	(−12.19 to −0.11)	58.54	(9.65 to 107.43)

TRB, T cell receptor β repertoires; TRG, T cell receptor γ repertoires; CI, confidence interval; ESR, erythrocyte sedimentation rate; CRP, C-reactive protein; RF, rheumatoid factor; ACCP, anti-cyclic citrullinated peptide; DAS28, disease activity score in 28 joints; HAQ, health assessment questionnaire; VAS, visual analog scale; SDAI, simplified disease activity index; CDAI, clinical disease activity index.

## Data Availability

Data available on request from the corresponding author.

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
