# Peer review of "Whole-Exome Sequencing and Analysis of the T Cell Receptor β and γ Repertoires in Rheumatoid Arthritis"

_diagnostics, 2024, doi:10.3390/diagnostics14050529_

Round 1

Reviewer 1 Report

Comments and Suggestions for Authors

I have no particular points. There are certain limitations such as the low number of patients and that changes in TCR repertoire cannot be linked to specific treatments/agents, nonetheless the results are interesting.

Author Response

The inclusion of a small number of RA patients and healthy controls is one of the limitations of this study, which is described in the ‘Discussion’ section. Thank you for your review.

Reviewer 2 Report

Comments and Suggestions for Authors

The authors investigated the clinical importance of T cell receptor repertoires for RA development in Korean patients.

Comments

1.      The paper requires significant English language improvement, primarily, word use. It also requires upgrading the logic and composition of the manuscript.

2.      Introduction is messy. It should be rewritten. All the repeated sentences should be corrected.

3.      Lines 43-44: This sentence should be removed.

4.       Line 117: Primer sequences should be presented.

5.      Line 159: The authors should describe the traits indicating that the identified WES are related to RA.

6.      Section 3.3: The authors should indicate which DMARDs were used for patients with RA treatment.

7.      Fig 1: The authors should indicate the difference (p values) between Controls and RA patients at all the examined time points.

8.      Table 4; Fig 2: the authors should indicate the number of patients used for correlation studies.

9.      Line 207, 223: It is not clear whether and why RA and HC subjects were combined for correlation studies. This should be clarified.

10.  Lines 242-262: These data is not clearly written and is related to Introduction and should be corrected and moved there.

11.  Lines 310, 317, 319 320: The TRB/TRG diversity ratio was not studied. This should be corrected.

Comments on the Quality of English Language

1.      The paper requires significant English language improvement, primarily, word use. It also requires upgrading the logic and composition of the manuscript.

2.      Introduction is messy. It should be rewritten. All the repeated sentences should be corrected.

1.      Lines 242-262: These data is not clearly written and is related to Introduction and should be corrected and moved there.

Author Response

Thank you for your review and advice.

Please refer to the attached file. Thank you.

Reviewer 3 Report

Comments and Suggestions for Authors

Jooyoung Cho et al report on T cell receptor repertoires in RA using whole exome sequencing.
Blood samples were taken from Normal and RA patients for WES, and clinical parameters were measured.
RA samples were collected prior to treatment and followed up at 6 and 12 months.
From whole blood, whole exome sequencing was complemented by targetting sequencing of the V and J regions of TCR genes.

Critical issues
>Selection criteria
There is a significant difference in age between RA patients and the healthy controls (p=0.034).  The average age of the HC is almost 20 years younger, and the interquartile ranges barely overlap.  The standard practice for selecting controls is to use age-matched controls to prevent age confounding the results.  Likewise, there is also a statistically significant difference in male:female ratio.  This is a critical failing of the study.
The authors should consider the inclusion of these Healthy donor data "for comparative purposes only", remove the statistical tests between controls in Figure 1, and note the shortcoming in the Discussion.
Please state if any RA pathogenesis SNPs were found in the healthy controls.

Major issues
>Methods
There is no description of the method for assessment of the Shannon-Wiener diversity.  There is no contextualisation of the TRB/TRG scores, which range from ~7 to ~10.  Can they be compared to other studies?

>Treatments and Response to treatment
The DMARDS used in the study are not specified. It would be helpful to have a list and a proportion of each treatment.  
Although there is an overall reduction in disease measures, it would be helpful to know if there were "non/poor-responders" and if their diversity differed from the responders.

>Figure 2
For Figure 2 it is assumed that all TSG/G data from all RA patients at all time points is plotted vs clinical parameters.
It would improve the figure if the data points were colour/symbol coded to denote Pre, 6m and 12m timepoints.

Minor issues
>Typos/language
There are a few typographical errors.  The five examples below are representative.
1 Ln 16 Shannon-Weiner -> Shannon-Wiener
2 Ln 272 PAID4 -> PADI4 or PAD4
3 Ln 278/280 TNF[alpha] vs TNFa.  There is a trend towards using just TNF.
4 Ln 280 TNF IL-6 IL-8.  Use italisized uppercase for official gene names.
5 Ln 39-41 "... identifying genes for RA is very important because it can be used as basic data in medicine as well as in public healthcare,...".  This sentence could be improved.

>Statistics
The authors should consider mentioning in their Methods/Statistical analysis section that statistical tests (apart from ANOVA) were made in isolation without accounting for multiple testing.

>Naive
The authors use the word "naive" to describe untreated patients.  Would "treatment-naive" be more precise?

Comments on the Quality of English Language

English is acceptable but would benefit from proof-reading.

Author Response

(The authors gave the same response as above.)

Round 2

Reviewer 2 Report

Comments and Suggestions for Authors

I have no more comments.